# Shoreline Dynamics and Evaluation of Cultural Heritage Sites on the Shores of Large Reservoirs: Kuibyshev Reservoir, Russian Federation

**Ionut Cristi Nicu** [1,*] , **Bulat Usmanov** [2] , **Iskander Gainullin** [3] **and Madina Galimova** [3]

1   High North Department, Norwegian Institute for Cultural Heritage Research (NIKU), Fram Centre, N-9296 Tromsø, Norway
2   Department of Landscape Ecology, Institute of Environmental Sciences, Kazan Federal University, 5 Tovarisheskaya Street, 420097 Kazan, Russia; busmanof@kpfu.ru
3   Khalikov Institute of Archaeology, 30 Butlerova Street, 420012 Kazan, Russia; gainullis@gmail.com (I.G.); mgalimova@yandex.ru (M.G.)
*   Correspondence: ionut.cristi.nicu@niku.no or nicucristi@gmail.com; Tel.: +47-98063607

**Abstract:** Over the last decades, the number of artificial reservoirs around the world has considerably increased. This leads to the formation of new shorelines, which are highly dynamic regarding erosion and deposition processes. The present work aims to assess the direct human action along the largest reservoir in Europe—Kuibyshev (Russian Federation) and to analyse threatened cultural heritage sites from the coastal area, with the help of historical maps, UAV (unmanned aerial vehicle), and topographic surveys. This approach is a necessity, due to the oscillating water level, local change of climate, and to the continuous increasing of natural hazards (in this case coastal erosion) all over the world. Many studies are approaching coastal areas of the seas and oceans, yet there are fewer studies regarding the inland coastal areas of large artificial reservoirs. Out of the total number of 1289 cultural heritage sites around the Kuibyshev reservoir, only 90 sites are not affected by the dam building; the rest had completely disappeared under the reservoir's water. The scenario of increasing and decreasing water level within the reservoir has shown the fact that there must be water oscillations greater than ±1 m in order to affect the cultural heritage sites. The results show that the coastal area is highly dynamic and that the complete destruction of the last remaining Palaeolithic site (Beganchik) from the shoreline of Kuibyshev reservoir is imminent, and immediate mitigation measures must be undertaken.

**Keywords:** cultural heritage; shoreline dynamics; GIS; UAV; Palaeolithic; Volga; European Russia

## 1. Introduction

The construction of large reservoirs along the large rivers of the world has, eventually, different effects: Local micro-climate modifications, disruption on the river flow regime [1], sediment transport [2,3], fauna [4], water chemistry [5], shore morphology [6–8], archaeology [9], fish yields [10], among other issues. They can also act as a place where different types of pollutants accumulate, and, in this way, it is easier to assess historical pollution [11]. One of the main effects is the triggering and the fast mechanic action of waves. These effects are accentuated by the global climatic changes, which are exponentially increasing every year.

Many studies deal with risk assessment [12,13], management [14,15], vulnerability [16–18], conservation strategies [19,20] and sustainability issues [21] regarding the cultural heritage of the coastal areas of seas and oceans. However, there is a lack of studies dealing with inland shorelines of large man-made reservoirs [22]. The Volga River is the largest river in Europe with a basin area of

1,360,000 km$^2$; it is considered the main river in Russia, and its basin represents the most significant economic region in Russia [23]. During the Soviet Union, there was a usual practice to flood large territories in order to obtain electricity and to relocate a large number of inhabitants and their houses. Unfortunately, cultural heritage sites do not enter this category; they cannot be relocated or moved.

During the Soviet period (the late 1930s), the "Great Volga Scheme" was initiated; the purpose was the construction of a chain of dams along the Volga River and one of its major tributaries—the Kama River. The reservoirs of the Volga-Kama cascade are one of the largest cascades in the world, totaling 11 reservoirs (Figure 1, Table 1). The main purpose of the dams was to produce electricity; before the 1930s, the Volga was used only for transport and fishing [24,25]. As shown in Table 1, Kuibyshev reservoir has the largest surface and the highest number of types of uses.

There have been limited studies referring to the destruction of archaeological sites around the Kuibyshev reservoir [26,27], but there are no studies referring to the entire surface of the reservoir. Therefore, this study is necessary to assess the exact number of sites impacted by the reservoir creation in 1957 and to draw attention for local authorities in their mission for future management plans [28] of the shoreline area [29]. A detailed case study was chosen to demonstrate the destructive potential of wave erosion; this was accomplished by a systematic monitoring process. The main scope of this article is (1) to track the major changes of the Volga River after the construction of the Kuibyshev reservoir with the help of GIS (2) to identify the area(s) that contain the highest concentration of archaeological sites (3) to analyse how many archaeological sites were impacted following the construction of the reservoir (4) to monitor the evolution of the only left Palaeolithic site—Beganchik from the shores of Kuibyshev reservoir, which has been specifically chosen because of its high erosion rates.

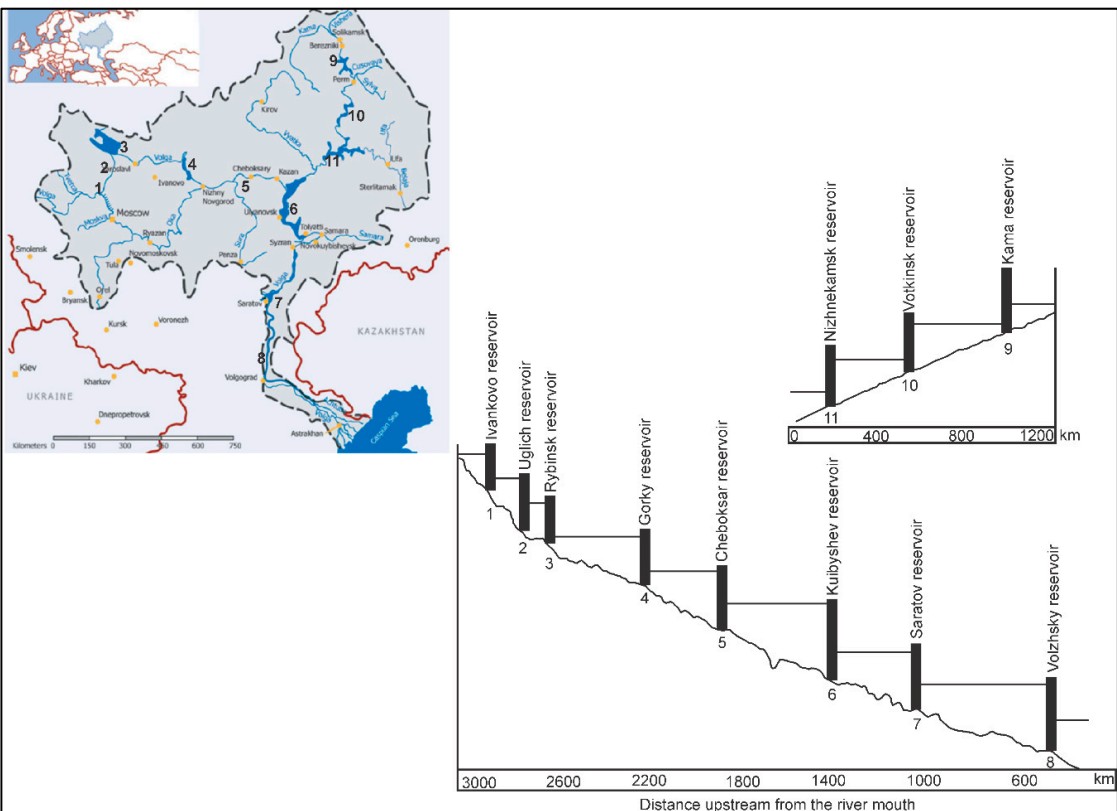

**Figure 1.** Detail of the reservoirs of the Volga-Kama cascade.

**Table 1.** The main characteristics of reservoirs from the Volga-Kama cascade [24,25] (the numbers in the first column correspond to the reservoirs from Figure 1).

| No. Crt. | YOC | RA (km$^2$) | Volume (km$^3$) | | IC (10$^3$ kW) | AO (10$^9$ kWh) | TOU |
|---|---|---|---|---|---|---|---|
| | | | Total | Useful | | | |
| 1 | 1937 | 327 | 1.2 | 1 | 30 | 0.12 | WNWrFPR |
| 2 | 1940 | 249 | 1.2 | 0.8 | 110 | 0.25 | PNWRF |
| 3 | 1941 | 4550 | 25.4 | 16.6 | 330 | 1.05 | PNWFFlWrRT |
| 4 | 1956 | 1770 | 8.7 | 2.8 | 520 | 1.4 | PNWFWrRT |
| 5 | 1981 | 3780 | 12.6 | 5.4 | 1404 | 3.3 | PNWRFWrT |
| 6 | 1958 | 6500 | 57.3 | 33.9 | 2300 | 10.2 | PNFIWFlWrRT |
| 7 | 1968 | 1950 | 12.8 | 1.7 | 1290 | 5.3 | PNWFlRTWr |
| 8 | 1960 | 3165 | 31.4 | 8.2 | 2530 | 10 | PNWFlFiWrRT |
| 9 | 1956 | 1845 | 12.2 | 9.8 | 504 | 1.7 | PTNFiWFRWr |
| 10 | 1961 | 1130 | 9.4 | 3.7 | 1000 | 2.2 | PNTWFiRWr |
| 11 | 1978 | 2305 | 13.8 | 4.6 | 1080 | 2.8 | PNTWFiRWr |

Legend: YOC—year of commissioning; RA—reservoir area; IC—installed capacity; AO—annual output; TOU—type of use; Fi—fishery, Fl—flood control, I—irrigation, Navigation, P—power production, R—recreation, T—timber rafting, W—water supply, Wr—water releases (sanitary, irrigation).

## 2. Study Area

Kuibyshev reservoir is a result of the construction of the Zhiguli Hydroelectric Station, Samara region, located between Zhigulevsk city (right bank of the Volga) and Tolyatti (left bank of the Volga); the reservoir covers the territory of regions Chuvash, Tatarstan, Ulyanovsk, and Samara. Kuibyshev reservoir has a surface of 6450 km$^2$, a volume of water of 58 km$^3$, a length of approximately 510 km, a mean depth of 9.3 m; these impressive numbers make it the largest reservoir in Europe [30], with a sedimentation rate of 8 mm/year. Important changes occurred in what concerns the sedimentation rate, which has fallen to 2.7–2.9 mm/year, after the commissioning of the dam in 1957. One of the main sources of sediments is a result of the abrasion processes, collapses of a huge amount of sediments into the reservoir [25,31].

Our area of interest is located in Tatarstan region (Figure 2a), on the left bank of Kuibyshev reservoir (Figure 2b), at the junction of Kama River in the Volga, about 75 km south-east of the city of Kazan (the capital city of Tatarstan). Beganchik site is located at approximately 2.8 km North-east of Izmeri village and 1.5 km north-west of Komintern village, on an isolated hill of the terrace above the floodplain; on the left bank of the confluence of Kama and Volga rivers, at the mouth of Aktai river (Figure 2c, Figure 3a).

The geology of the area consists of Permian, Pliocene and Quaternary deposits. Quaternary sediments are dominant in Volga-Kama terraces, eight palaeohydrological phases were identified from high and low fluvial activity; the most recent active phase corresponds to the Little Ice Age [32]. There is a limited number of studies regarding the evolution of the coastal area in the Tatarstan region, Russia [25], along with the analysis of landslides [33] and gully erosion [34].

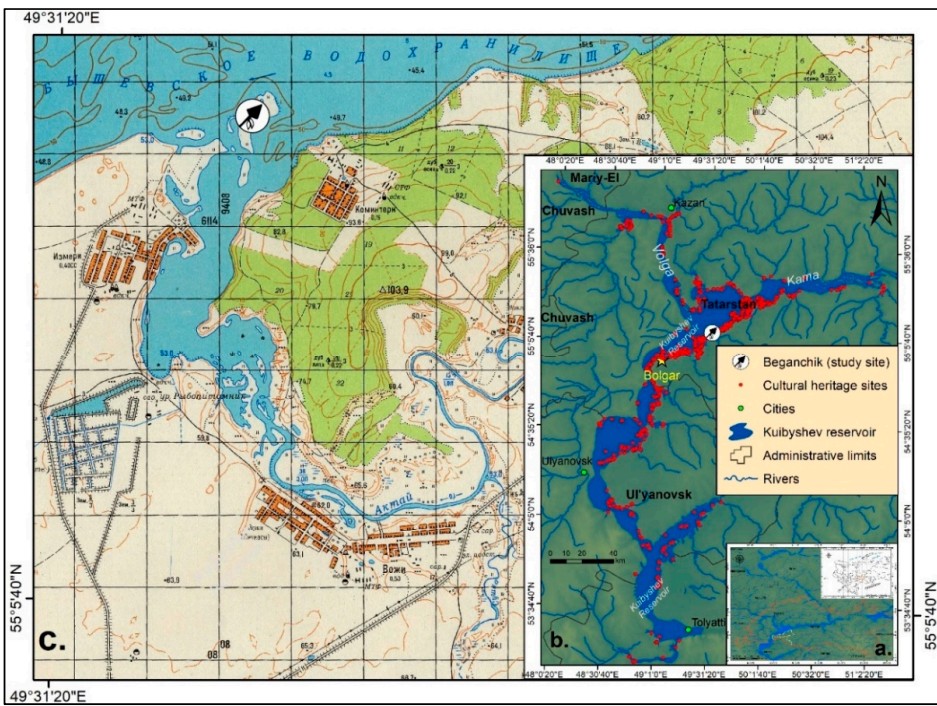

**Figure 2.** Geographical location of: (**a**) Kuibyshev reservoir in a regional context; (**b**) Kuibyshev reservoir and the cultural heritage sites around it; (**c**) Beganchik site on the topographic maps scale 1:50.000 (edition 1984).

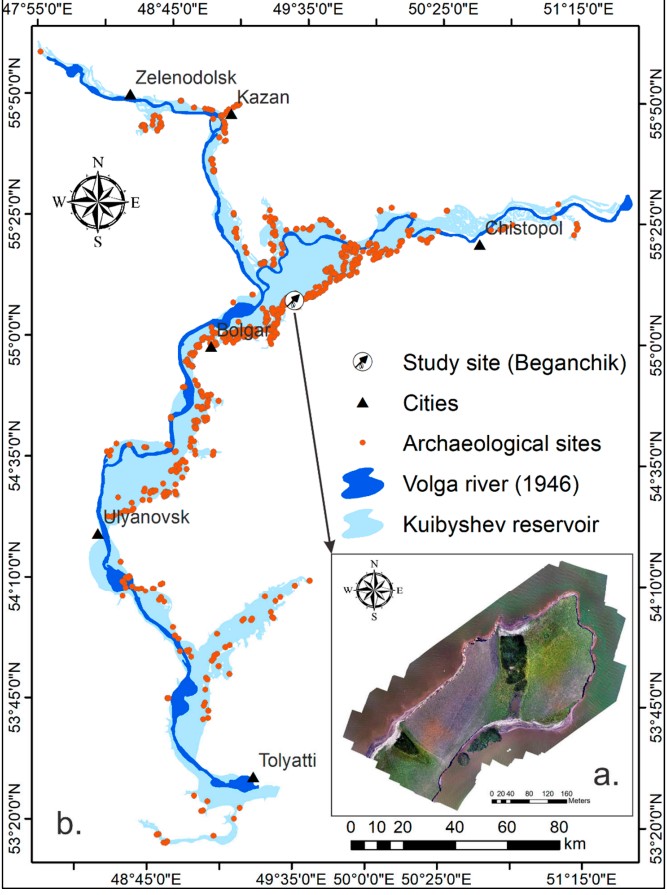

**Figure 3.** (**a**) Aerial image of Beganchik site from 2018; (**b**) Overlapping Volga River before and after the building of Kuibyshev reservoir, along with the inventory of archaeological sites.

## 3. Archaeological Background

### 3.1. General Overview

Ever since the Palaeolithic, large rivers and their fertile plains have been a magnet for prehistoric people to place their settlements; water is undoubtedly the most important resource that a community of people needs in order to decide where to place a settlement [35]. According to the local geographical factors and paleogeographic evolution of the landscape, the old location could be used by the next population of a different historic period. This is how multi-stratified archaeological sites were created. Around Kuibyshev reservoir, 1289 cultural heritage sites have been identified. The protection of cultural heritage assets in Russia was and is still a current issue, because of the country's complex political background; at present, cultural heritage is protected by the Federal Law 73-F3, On the Objects of Cultural Heritage (Historical and Cultural Sites) of the Peoples of the Russian Federation [36].

In this area, the oldest traces are attributed to Upper Palaeolithic-Mesolithic period. The area surrounding the Volga River has tremendous potential in regard to cultural heritage sites of Palaeolithic [37,38], Mesolithic age [39], Neolithic [40], Chalcolithic/Bronze Age [41], Early Iron Age [42], Middle Ages, etc. The only remaining Palaeolithic site which has not been impacted by the reservoir is Beganchik. Besides the Palaeolithic site, Beganchik, around the Kuibyshev reservoir there are many archaeological sites of international and national significance; among them, the Bolgar archaeological site. The Bolgar Historical and Archaeological Complex are part of the UNESCO World Heritage List since 2014; it represents the existence of the Volga-Bolgar civilisation (7–15th centuries AD), and the first capital of the Golden Horde in the 13th century [25].

### 3.2. Beganchik Site

Beganchik site was studied for the first time in September 1985 by M. Sh. Galimova and K. E. Istomin at the recommendation of E.P. Kazakov; the first description of the site is the islet named "the Izmeri Island". In 1981, mammoth fauna fossils were found; they were located on the towing-path in the south-western part of the islet, at the foot of the narrow, long butte, which had the shape of a peninsula with a length of about 200 m. The discovered mammoth fossils were five teeth and leg bones, together with large flint nuclei and tools in an area of $20 \times 20$ m$^2$. Unfortunately, by the year 2000, this peninsula was completely eroded by the Kuibyshev reservoir. In the next years, almost every autumn (until 2012) Kazakov collected stone artefacts and faunal remains at the south-western tip of the islet in conditions of low reservoir level [43].

M. Sh. Galimova in 1985–1987 and 2000 also conducted investigations of this site. In 1986, a reference point was installed at the highest point of the islet, therefore all excavations and trenches were referenced after it. An excavation area of 104 m$^2$ was located on the edge of the steep west coast of the islet; the cultural layer has been found at 100–130 cm depth; 1968 of artefacts were found. The specific features of the Beganchik stone industry, which was based on blade production by means of striking technique and its flint inventory, allowed M. Sh. Galimova to frame the site of initial (Upper Palaeolithic) period of the Mesolithic Ust-Kama culture. The main diagnostic tool of the Ust-Kama inventory is the arrowhead in a trapezoid shape with concave sides, which were shaped by retouching [44]. In 2000, the rescue excavations of the site were continued by M. Sh. Galimova with the participation of I. I. Gainullin. By that moment excavation territory of 1986–1987 was eroded by the reservoir. In general, the western and northern coasts of the islet were washed away by 20–25 m from the erosion ledge for 14 years (1986–2000). In the autumn of 2012, rescue investigation of the Beganchik site and Izmeri I site was conducted by the expedition of the National Centre for Archaeological Research of the Tatarstan Academy of Sciences. In the autumn of 2013, rescue investigation on the Beganchik islet was continued by a joint campaign of the "Expedition for Prehistory" of the Institute of Archaeology of the Tatarstan Academy of Sciences and "Archaeological Expedition" of the Chuvash State Institute for Humanities. The total excavated area was 20 m$^2$; following the excavation and

surface findings, a significant collection of stone artefacts (439 items) and 80 bones of a mammoth were found [45].

## 4. Materials and Methods

In order to determine the shoreline dynamics, topographic map scale 1:300,000 (edition 1945), and Google Earth images from 2010, were employed. From the topographic map, the extent of the Volga River before the reservoir construction was digitised; from Google Earth images from 2010, the extent of the reservoir was digitised. They were overlapped in ArcGIS and the highest differences were observed.

The archaeological inventory, as a point feature, (Figure 3b) was provided by the Institute of Archaeology of Tatarstan Academy of Sciences. The database was compiled over a long period of time, both prior and after the filling of the reservoir; first sites were described in the early 1940s until the early 1960s, when special survey expeditions were undertaken, with the aim to find as many sites and record brief information about them. After the filling of the reservoir, more expeditions were undertaken to highlight the impact on the sites. Other sources in building the database included the descriptions of the Archaeological Maps of Tatarstan, Ulyanovsk and Samara regions. A survey has been unsystematic across the study area, sites are located to varying degrees of accuracy and the full extent of individual sites is not necessarily known. The database is still under construction, as the area is very large and only a few people within the Institute of Archaeology of Tatarstan Academy of Sciences is working to continuously update it. However, it is the most comprehensive archaeological dataset which currently exists in this area, hence will be used for this study. Density analysis of the settlements for the main historical periods was performed; this was made using the Point Density feature (using the circle as neighbourhood option) from ArcToolbox (ArcGIS).

The danger towards increasing and decreasing water level over the digital elevation model (DEM) was evaluated by making four working scenarios; the DEM used in this study is based on the Shuttle Radar Topography Mission (SRTM), with a pixel size of $30 \times 30$ m$^2$. First, the water level was decreased by 0.5 m and 1 m, followed by increasing the water level with 0.5 m and 1 m, respectively. Changes in reservoir level regime occur out of two main reasons: Natural seasonal changes in the flow and artificial regulations of water discharge through hydraulic structures, the difference in baric pressure, wind speed and changes in the hydraulic slope. The water level of the reservoir is controlled by a special department—RusHydro. We chose these values taking into consideration our large-scale study area and to point out the minor oscillations in water level by arbitrarily increasing/decreasing it by $\pm 0.5$ to 1 m. In order to have a better image of the changes occurred along the Volga River after the Kuibyshev reservoir was built, the entire area was divided into three sectors, as follows: Sector 1 (Figure 4b), Sector 2 (Figure 4c), and Sector 3 (Figure 4d).

For monitoring the Beganchik site, a rich cartographic background, including Soviet aerial images from 1958 and 1980, Roscosmos aerial images (2008), and UAV flights [46] from the summer of 2017 and 2018 were used. As shown in numerous studies, old maps and aerial images are an important source of information [47] that can be easily digitised, integrated into GIS [48], and used in the field of cultural heritage [49,50]. All the maps and aerial photos have been georeferenced with the help of ArcGIS. The ground control points used for the drone flights were measured with a GNSS (Global Navigation Satellite System) receiver Trimble Geoexplorer 6000 XH and Leica Zeno 20. The UAV is a drone, model DJI Phantom 4. The survey was performed with a 12-megapixel camera mounted on the quadcopter; the UAV was controlled from a smartphone using Pix4D Capture software, which allows configuring the shooting parameters. Aerial photography was performed with the following parameters, height—70 m, picture overlapping—60–80%, camera position—90 degrees, meteorological conditions—no precipitation, and wind no more than 15 m/s. The photos were processed using the algorithms built into the Agisoft Photoscan software; the resulting model was processed by a polynomial approximation exponential kernel (PAEK) method with 1 m tolerance.

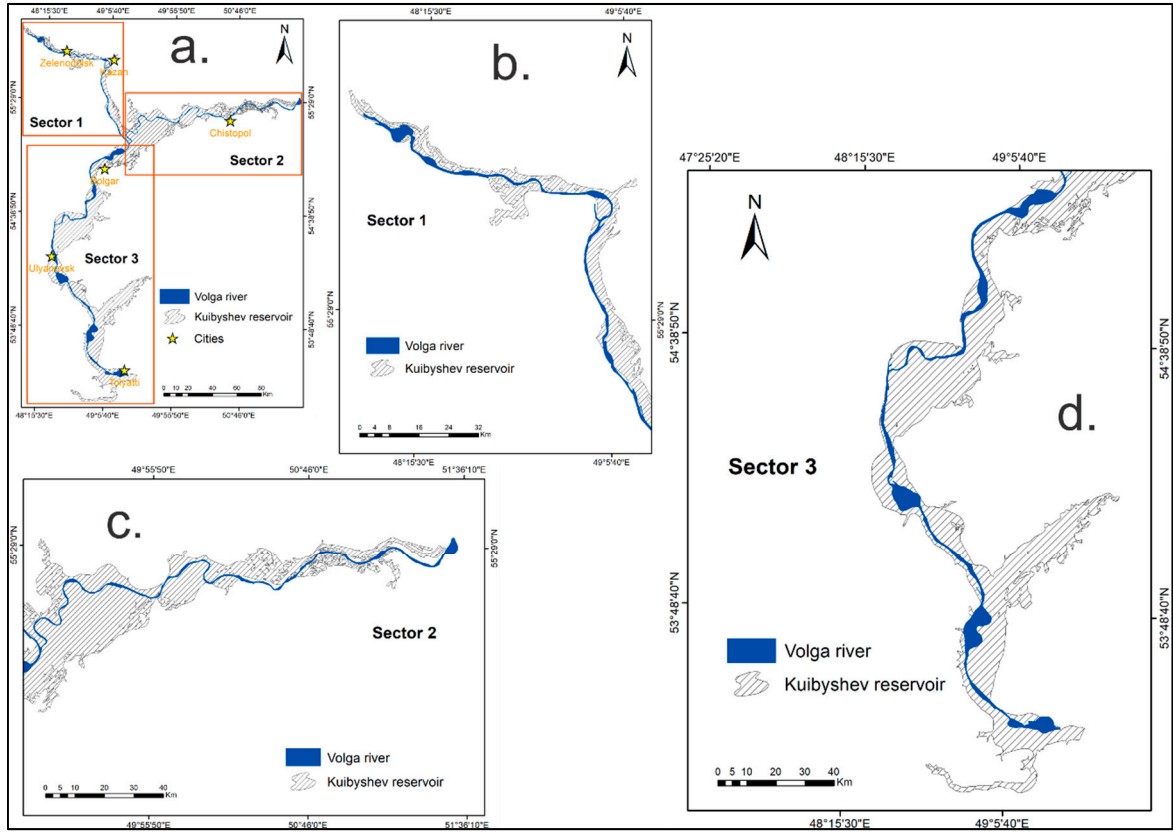

**Figure 4.** (**a**) General division of the Kuibyshev reservoir; (**b**) Sector 1; (**c**) Sector 2; (**d**) Sector 3.

## 5. Results

Each sector will be analysed according to the GIS integration of the spatial data collected from old maps and modern aerial images, followed by the analysis of the archaeological sites patterns and dynamics along the Volga River; then, the changes of Volga River will be analysed in the context of how many cultural heritage sites are directly affected by the reservoir construction. The four working scenarios will be analysed in order to evaluate the endangered sites towards increasing/decreasing water level of the reservoir. Finally, the monitoring results of the only left Palaeolithic site—Beganchik will be presented; this site has been specifically chosen because of its high erosion rates and being the only remaining Palaeolithic site around Kuibyshev reservoir.

### 5.1. Volga River Dynamics

From the town of Tver to Volgograd, Volga River flow velocity is affected by the 8 reservoirs. The reservoirs were built to control seasonal changes in flow; however, there are no significant changes when it comes to river discharge and the total annual discharge. In the middle Volga, the mean annual flow from 1876 to 1940 was 2876 m$^3$/s; after the construction of reservoirs, from 1942–1955, the mean annual flow was 2780 m$^3$/s [51].

**Sector 1** (Figure 5a) stretches approximately in the north-western part, next to the Zvenigovo city till south-east, at the junction between Volga and Kama rivers; it has a length of approximately 145 km. The most important cities within Sector 1 are Kazan (with a population of about 1,2 mil. people) and Zelenodolsk (with a population of about 98,000 people). Out of the three sectors, Sector 1 has the fewest changes, compared with the others; the most significant changes are located about 43 km downstream from Kazan city. Initially, Volga had a width of 1.4 km, while after the building of the Kuibyshev reservoir the width of Volga reached 9.5 km. Another significant change is located between Zelenodolsk and Kazan, at the junction of Sviyaga River in the Volga; from a width of 0.6 km, Volga reached a width of 11.2 km; except this, the reservoir water has mainly covered the left side of



the river. This is due to the geomorphological characteristics of the area; the right side represents the Volga uplands, while on the left side are the terraced plains of the lowland Volga River region [52].

**Sector 2** (Figure 5b) stretches from east to west on a distance of approximately 150 km and represents the lower Kama River junction to Volga River; before the Kuibyshev reservoir, the area located around the junction had a significant number of villages (which were completely destroyed). Moreover, important landscape changes occurred, along with an acceleration of coastal erosion with a direct effect on cultural heritage [25]. The most important city in this sector is Chistopol (with a population of approximately 60,000 people). Along its approximately 150 km length, after the building of Kuibyshev reservoir, Sector 2 had more or less a balanced development of the right and left bank; this is because both of the sides are located within the terraced plains of the lowland Volga River region. From an average width of 0.8–1 km, the Kama River reached widths of 13–36.8 km. From these numbers, we can realise the real proportions of the consequences of the Kuibyshev reservoir being built.

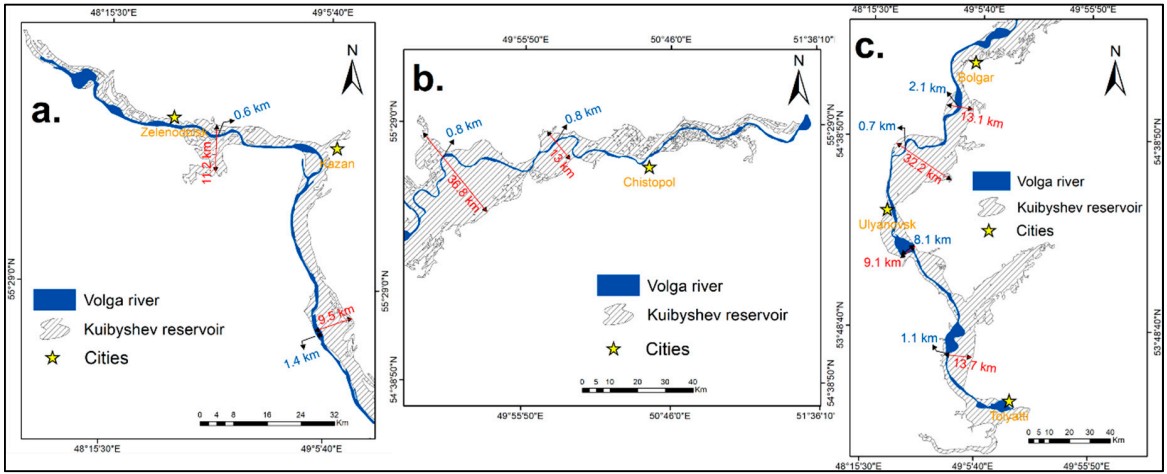

**Figure 5.** Results of the coastal dynamics analysis for: (**a**) Sector 1; (**b**) Sector 2; (**c**) Sector 3.

**Sector 3** (Figure 5c), with a length of approximately 263 km, stretches from Kama and Volga junction until the dam of the Zhiguli Hydroelectric Station, located between the cities of Zhigulyovsk and Tolyatti. The most important cities in this sector are Tolyatti (with a population of approximately 720,000 people), Ulyanovsk (with a population of approximately 614,000 people) and Bolgar (with a population of approximately 9000 people). Bolgar is well known for the Bolgar Historical and Archaeological Complex World Heritage site. Similar in development with Sector 1, the left side of the river being more developed than the right one; again, this is to the lower altitudes of the terraced plains of the lowland Volga River region [52]. Within this sector, the width of the Volga River before Kuibyshev reservoir was ranging from 0.7–2.1 km, and from 9.1–32.2 km after the building of Kuibyshev reservoir. Having this enormous width, it is sometimes called the Kuibyshev Sea.

*5.2. Archaeological Site Analysis*

Following the analysis of the archaeological database provided by the Institute of Archaeology of Tatarstan Academy of Sciences, the following periods were identified: Palaeolithic/Mesolithic, Neolithic, Chalcolithic/Bronze Age, Early Iron Age, Migration Period, and Middle Ages. Large river systems, e.g., the Volga, act as a magnet when it comes to taking a decision to place a prehistoric settlement. That is why, in the close proximity of the Volga River and its tributaries, there is a high density of archaeological sites. Water represents the main resource in establishing the placement of a settlement; this is documented and well-known across the archaeologists and geo-archaeologists [53].

On the basis of the existing database, the areas with the highest concentration of archaeological settlements attributed to a certain period will be identified and highlighted accordingly. Usually, the dynamics of the settlements are influenced by different factors, like climate change [54], natural

hazards [55] and threats from other populations. Having knowledge of the spatiotemporal distribution patterns of archaeological sites is a powerful tool to understand past human-environment interactions and to evaluate landscape vulnerability to natural [56] and anthropogenic changes [49].

　　　As can be seen in Figure 6, the highest concentration of settlements for all the periods is located at the junction of the Kama and Volga Rivers, this is representing an important communication route. During the Palaeolithic/Mesolithic period (Figure 6a), the hunters-fishers-gatherers population was well adapted to the living around water bodies and forests; the highest concentration was at the junction of Kama and Volga rivers, followed by the adjacent areas of upstream and downstream of the junction. A good concentration can be observed on the Volga River, around the area where presently the city of Kazan is located; the thrive of settlements was due to the optimum climatic conditions for the Preboreal period [37,39], along with the highest levels of rivers and lakes, which was typical for the Mesolithic epoch [57]. The settlements were not very homogenous during this period. However, this can be observed during the Neolithic period (Figure 6b), when more settlements appear in the area between the today Kazan city and Kama-Volga junction. The Neolithic period is characterised by the emergence of pottery, new types of stone tools and the transition to sedentism with the help of active fishing and hunting. The majority of Neolithic sites are located on the remnants of the floodplain of the small rivers of the Kama River tributaries or on the first terrace of the Kama River [40].

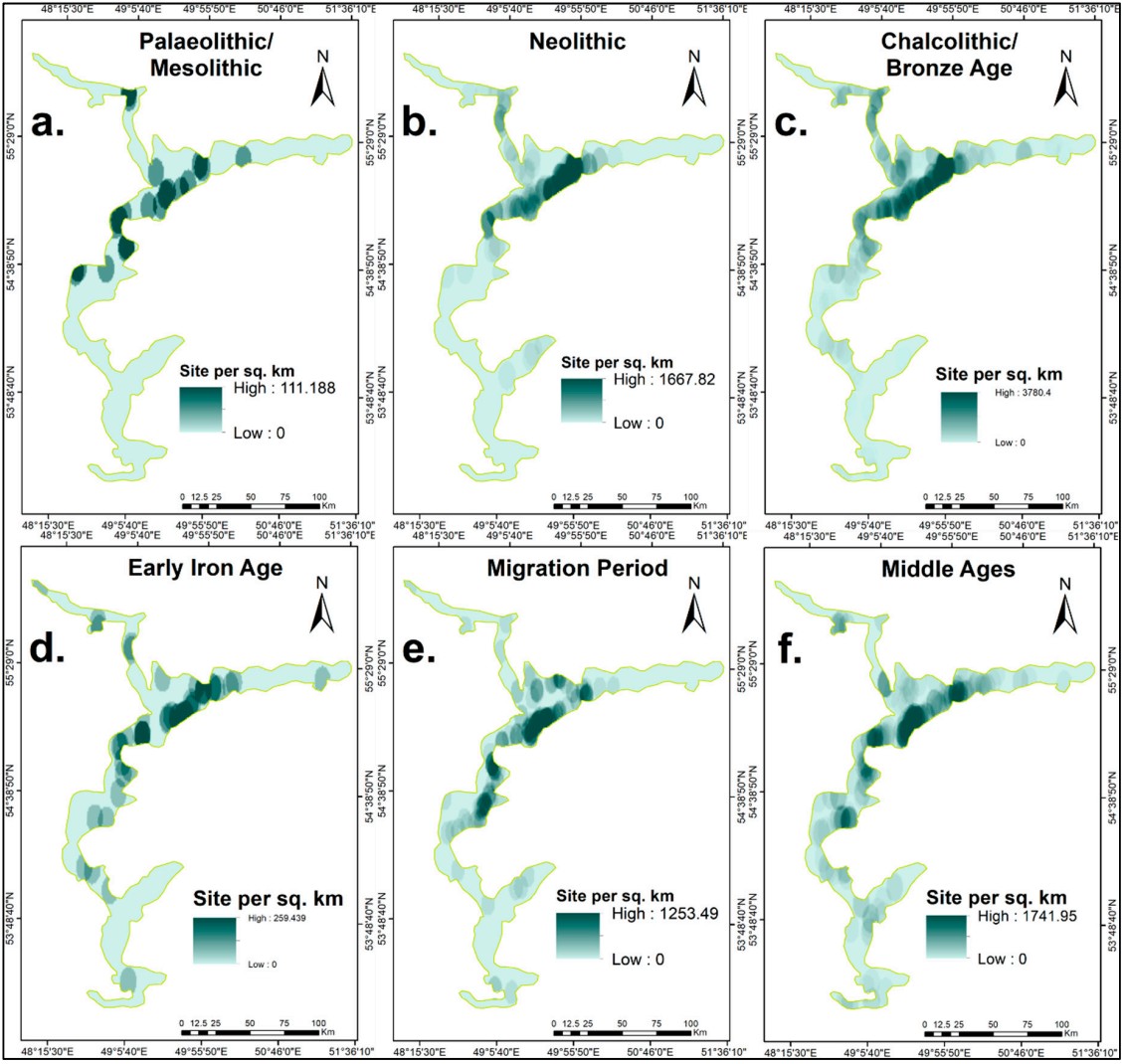

**Figure 6.** Location and density analysis for archaeological sites for the following periods: (**a**) Palaeolithic/Mesolithic; (**b**) Neolithic; (**c**) Chalcolithic/Bronze Age; (**d**) Early Iron Age; (**e**) Migration Period; (**f**) Middle Ages.

Following Chalcolithic/Bronze Age period (Figure 6c), it can be observed even a higher degree of homogeneity among the settlements; this is due to the fact that the lowest levels of water were recorded in the Bronze Age. As a consequence of this, even the lowest altitudes were chosen to place the settlements, which is why the analysis shows a larger continuous surface

Figure 6d illustrates the density of the Early Iron Age settlements, which started to be more fragmented. The highest concentration is at the confluence of Kama-Volga Rivers and on the territory of today Bolgar, followed by scattered low-density areas the upstream Volga, at the mouth of Sviyaga River, and the downstream Volga. The settlements appear scattered because of their higher altitudinal position (higher position throughout the Holocene), due to the associated high flood levels [57]. As can be seen in Figure 6e, the Migration period is characterised by spreading of population downstream Volga River, until today Ulyanovsk city. However, the highest concentration is still located at the Kama-Volga junction; the fact that during this period the area is very poorly populated is also indicated by [58]. Finally, the Middle Ages (Figure 6f) show the highest fragmentation of the settlements. The highest concentration remains the same (Kama-Volga junction), while the settlements are scattered downstream and the upstream Volga until today Tolyatti and Zelenodolsk, respectively. During this period, the settlements are so scattered, due to the fact that the climatic conditions were suitable for the long-term occupation of river and lake floodplains. This is the turning point when people start to settle and make semi-permanent settlements and start off using the floodplain in order to practice agriculture on a higher level [57,59].

### 5.3. Cultural Heritage under Erosion Threat

Since the formation of the reservoir in the middle of the 1950s, the confluence of the Kama and Volga Rivers and the left bank tributaries was flooded. As a result, many lower terraces, that were hosting archaeological sites of different periods [60], were completely flooded. The main typology of the sites is presented in Table 2; therefore, out of the total of 1289 sites, 1091 are underwater or totally impacted following the building of the Kuibyshev reservoir. According to their chronology, shown in Table 3, the only Palaeolithic/Mesolithic site that still exists, but is under high threat from coastal erosion, will be further analysed, based on the old Soviet Maps and modern surveys.

Based on the working scenarios regarding the water level increasing and decreasing to 0.5 m and 1 m, respectively it has been observed that increasing the water level, whether, with 0.5 or 1 m, a number of two extra sites will be affected (out of 1091 already underwater or impacted). If we decrease the water level by 0.5 m or 1 m respectively, the same number of sites will remain affected—1091. Having such a large surface, water level oscillations do not affect the cultural heritage sites, unless there are variations greater than ±1 m.

### 5.4. Beganchik Site

In order to analyse the coastal dynamics of Beganchik site, all the surveys were overlapped, and the site was divided into three sectors (Figure 7), which will be further analysed separately. Beganchik site is located at the mouth of Aktai River, on the second terrace (the first terrace being flooded by Kuibyshev reservoir) of the floodplain which formed before the Holocene [61]; the altitude is between 54–60 m a.s.l. According to the general view of the site (Figure 8a), the northern part of the site is represented by a very steep cliff (Figure 8b) which is continually eroding. Previous preliminary studies [44] have revealed that the erosion rate is about 2–3 m/year.

**Table 2.** Distribution of cultural heritage sites around Kuibyshev reservoir according to their typology.

| Type | Building | Burial Ground | Burial Mound (s) | Complex Site | Fortified Settlement/HILLFORT | Hoard | Surface Find | Tombstone | Unfortified Settlement | Total |
|---|---|---|---|---|---|---|---|---|---|---|
| Number | 2 | 103 | 25 | 4 | 40 | 17 | 179 | 4 | 915 | 1289 |
| Affected (under water) | 0 | 82 | 21 | 3 | 31 | 14 | 156 | 4 | 780 | 1091 |
| Not affected | 2 | 21 | 4 | 1 | 9 | 3 | 23 | 0 | 135 | 198 |

**Table 3.** Distribution of cultural heritage sites around Kuibyshev reservoir according to their chronology.

| Age | Chalcolithic/Bronze Age | Early Iron Age | Middle Ages | Migration Period | Modern Time | Neolithic | Palaeolithic/Mesolithic | Not Identified | Total |
|---|---|---|---|---|---|---|---|---|---|
| Number | 566 | 45 | 275 | 148 | 11 | 164 | 20 | 60 | 1289 |
| Affected (under water) | 490 | 39 | 223 | 118 | 6 | 144 | 19 | 52 | 1091 |
| Not affected | 76 | 6 | 52 | 30 | 5 | 20 | 1 | 8 | 198 |

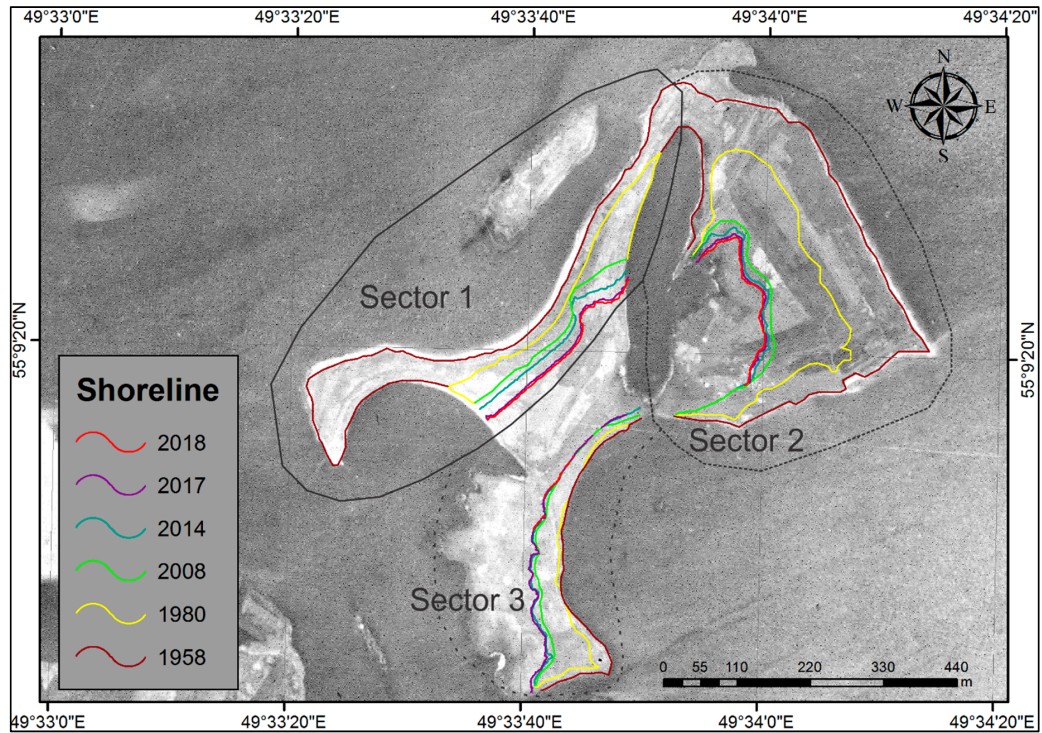

**Figure 7.** Shoreline limit resulted from cartographic analysis and field surveys and the division of the three analysed sectors.

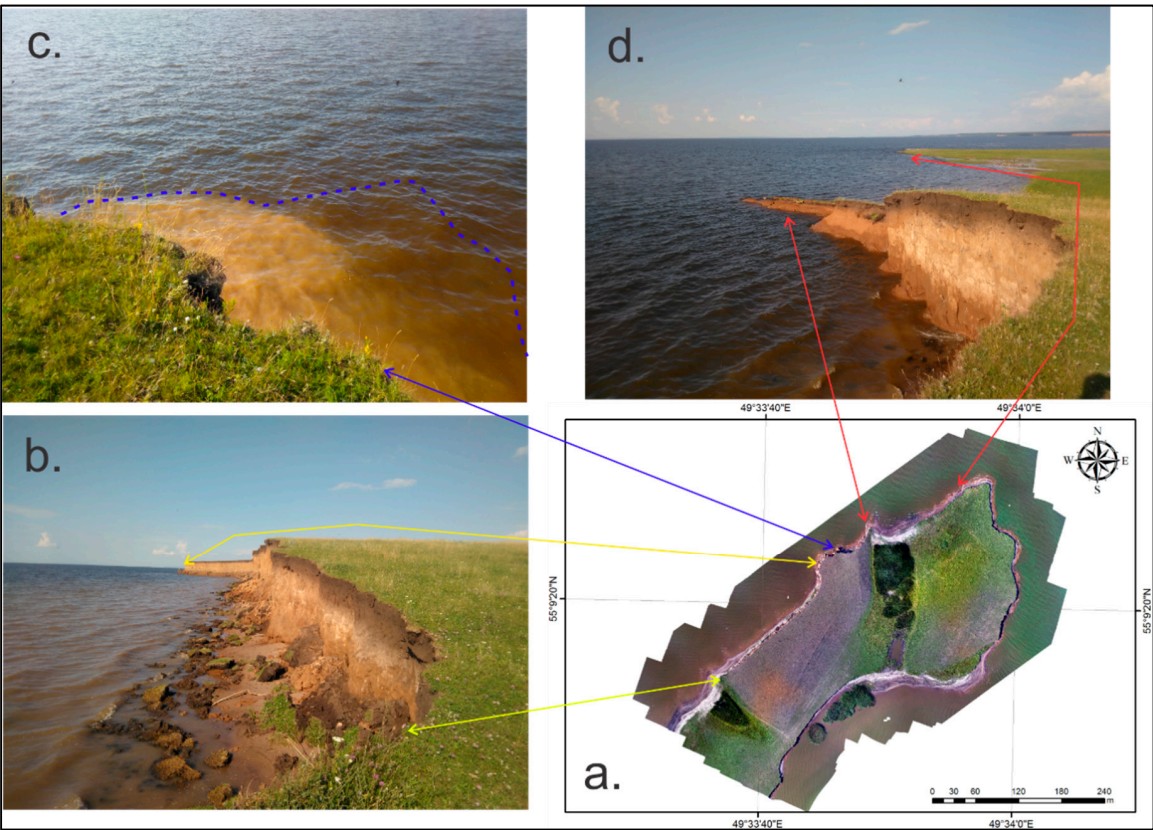

**Figure 8.** (**a**) General view of Beganchik site (drone flight) from 2017; (**b**) Detail over the northern part of the site, fresh parts from the coast are visible in the water (August 2017); (**c**) The change of water colour, due to the clay content of the soil; (**d**) The northern part of the site, where the height of the coast is decreasing.

### 5.4.1. Sector 1

Sector 1 was not actively eroded between 1958 and 1980 because it was protected by another island (60–90 m north-west, Figure 7), as indicated by the relatively low values of the shoreline retreat (Table 4); in this way, the site was protected from the mechanical action of waves (Figure 8c). Later on, it can be seen that after the island disappeared, the yearly erosion has considerably increased, along with the specific land loss and volume. The direction of the Kama River flow is from north, north-east; being located at the "shelter" of Sector 2 from the speed and currents of the Kama River, this section was in some way protected. However, this sector became likely to be eroded, due to the high erosion rates of Sector 2 and having an elongated shape; this is highlighted of the specific land loss for 1958–1980.

**Table 4.** Detailed morphometric indicators from different observation periods for Sector 1.

| Observation Period | Years | Shoreline Retreat | | Eroded Area | | Specific Land Loss | Specific Volume Loss |
|---|---|---|---|---|---|---|---|
| | | m | m/year | ha | ha/year | $n * 10^{-3}$ ha/km * year | thousands $m^3$/km * year |
| 1958–1980 | 22 | 32.84 | 1.5 | 2.77 | 0.13 | 253.13 | 12.65 |
| 1980–2008 | 28 | 33.88 | 1.21 | 1.26 | 0.04 | 133.48 | 6.67 |
| 2008–2014 | 6 | 11.57 | 1.93 | 0.39 | 0.06 | 191.87 | 9.59 |
| 2014–2017 | 3 | 17.03 | 5.68 | 0.50 | 0.17 | 469.53 | 23.48 |
| 2017–2018 | 1 | 3.23 | 3.23 | 0.09 | 0.09 | 276.55 | 13.83 |

Note: * defines multiplication.

Very high values of the specific land loss, in comparison with other sectors, is due to the height of the coast; which, in some parts can reach 5 m in height. Following the analysis, Sector 1 can be characterised as an extremely dangerous one.

### 5.4.2. Sector 2

Unlike Sector 1, Sector 2 was and still is under the direct exposure of the Kama River flow and currents. As can be seen in Table 5, the specific land loss is at extremely high rates (Figure 8d). This sector is the most exposed and threatened by erosion. The shoreline retreat is generally stable, varying within 2 m. According to the specific land loss indicator, it can be observed that the destruction occurred, especially within the first two periods, which is typical for the initial stage of lowland reservoir development. During this period, the extremities of this sector are cut off, after which the erosion process stabilises. Between 1958–2008, approximately 70% of the eastern part of the site was eroded. Following that, part of the river's current's strength was redistributed along the north-western part, which explains the sudden decrease in land loss. The height of the coast does not exceed 2 m, therefore, Sector 2 can be classified as moderately dangerous.

**Table 5.** Detailed morphometric indicators from different observation periods for Sector 2.

| Observation Period | Years | Shoreline Retreat | | Eroded Area | | Specific Land Loss | Specific Volume Loss |
|---|---|---|---|---|---|---|---|
| | | m | m/year | ha | ha/year | $n * 10^{-3}$ ha/km * year | thousands $m^3$/km * year |
| 1958–1980 | 22 | 44.85 | 2.04 | 4.68 | 0.21 | 279.75 | 4.2 |
| 1980–2008 | 28 | 64.81 | 2.31 | 3.51 | 0.13 | 288.84 | 4.33 |
| 2008–2014 | 6 | 9.61 | 1.6 | 0.27 | 0.04 | 89.91 | 1.57 |
| 2014–2017 | 3 | 4.76 | 1.59 | 0.13 | 0.04 | 82.24 | 1.44 |
| 2017–2018 | 1 | 2.8 | 2.8 | 0.08 | 0.08 | 189.95 | 3.32 |

Note: * defines multiplication.

### 5.4.3. Sector 3

Sector 3 is located in the close proximity to Aktai River mouth, where is protected from the mechanical action of waves and Kama River strong currents. This portion of the Beganchik site shoreline is the most stable. As it can be seen from Table 6, there have been no significant changes

regarding this part of the coast from 1958 to 2018; except the period 1980–2014, when the specific land loss is higher when compared to other periods, but considerably lower when compared with the other two sectors. The most intensive processes of coastal transformation in the study area were observed in Sectors 1 and 2, open to the destructive effect of the currents and the mechanic action of waves. The erosion intensity may vary from year to year, depending on the water level oscillations in the reservoir. In order to have a more detailed situation on the Beganchik site erosion rates, continuous annual observations are needed.

**Table 6.** Detailed morphometric indicators from different observation periods for Sector 3.

| Observation Period | Years | Shoreline Retreat | | Eroded Area | | Specific Land Loss | Specific Volume Loss |
|---|---|---|---|---|---|---|---|
| | | m | m/year | ha | ha/year | n * $10^{-3}$ ha/km * year | thousands $m^3$/km * year |
| **1958–1980** | 22 | 5.59 | 0.25 | 0.06 | - | 16.55 | 0.25 |
| 1980–2008 | 28 | 13.85 | 0.49 | 0.22 | 0.01 | 47.49 | 0.71 |
| 2008–2014 | 6 | 7.37 | 1.23 | 0.05 | 0.01 | 46.24 | 0.69 |
| 2014–2017 | 3 | 1.53 | 0.51 | 0.01 | - | 14.17 | 0.21 |
| 2017–2018 | 1 | 0.31 | 0.31 | 0.002 | 0.002 | 9.11 | 0.14 |

Note: * defines multiplication.

Particular attention should be paid to Sector 1, in which the most important part of the site is located. If the erosion rates remain stable, the site will be completely impacted in about two or three decades. This imposes urgent mitigation measures from local authorities, along with the sustainable management of cultural heritage sites.

## 6. Discussions

Reservoir construction had a significant impact on the flow regime because the current velocity decreased. The currents are very complex, as river flows are under the direct effect of convective flows and wind effects formed in the reservoirs. These are characteristic for Kuibyshev reservoir, where wind effect and bottom relief have a high influence on hydrological conditions [51]. Volga River frames itself into the future increase of global river flow as a consequence of climate change; predictions have shown an increase of 4–8% during 2071–2100 [62]. To be more specific, future trends in the area show an increase in precipitation, temperature and in the use and levels of waters in rivers [63]. The cyclic oscillations have occurred in the Volga River basin in the last half-century; this has influenced the water level in the reservoir, and therefore the erosion rates of the shoreline. The numbers in the Tables 4 and 5 are related to the cyclic oscillations that brought two high-water periods (1951–1962, 1977–1995) and two low-water periods (1963–1976, 1996–present) [51]. For Sectors 1 and 2 high-water levels are associated with low erosion, while the low-water level is associated with higher erosion rates. The research presented in this paper continues our endeavour to monitor the endangered cultural heritage sites from the shoreline of the Kuibyshev reservoir [25–27,60]. Combining old maps with new data collected from field surveys shows high efficiency in establishing the erosion rates of archaeological sites located on shorelines of big reservoirs. When comparing erosion rates with the previous study [25], the average shoreline retreat is close (~3–4 m/year).

## 7. Conclusions

In this study, the main changes along the largest reservoir in Europe—Kuibyshev (Russian Federation) were analysed, in strong connection to cultural heritage sites. Following the analysis, Sector 2 has been identified as one with the highest values of width oscillation, from 0.8–1 km to 13–36.8 km. Cultural heritage sites located in the close proximity to big rivers and/or big reservoirs are especially subjected to erosion from water, water level oscillations, and the mechanical action of waves. A diachronic analysis of the archaeological sites located along the Volga River and its main tributaries has highlighted the fact that the most inhabited area was located at the junction of Kama River into the Volga. As highlighted in our analysis, 85% of the cultural heritage around Kuibyshev

reservoir is impacted. However, a more thorough process of monitoring and evaluating the present state of cultural heritage is needed. This has to be done with the cooperation of local authorities and stakeholders. The survey of the only left Palaeolithic site—Beganchik, has shown a fast degradation with no mitigation measures from the local authorities. Beganchik site remains promising for regular rescue archaeological excavations, despite the loss of more than a half of its surface, due to coastal erosion over the last 30 years. Working on scenarios regarding the management of archaeological sites around Kuibyshev reservoir represents one of our future goals.

**Author Contributions:** Conceptualisation, I.C.N. and B.U.; methodology, I.C.N., B.U. and I.G.; software, I.C.N., B.U. and I.G.; formal analysis, I.C.N. and B.U.; investigation, I.C.N., B.U. and I.G.; resources, I.C.N., B.U., I.G. and M.G.; data curation, I.C.N., B.U., I.G. and M.G.; writing—original draft preparation, I.C.N. and B.U.; writing—review and editing, I.C.N., B.U., I.G. and M.G.; visualisation, I.C.N. and B.U.; supervision, I.C.N. and B.U.; review corrections, I.C.N. and B.U.

**Funding:** This work is performed according to the Russian Government Program of Competitive Growth of Kazan Federal University.

**Acknowledgments:** The archaeological database was kindly provided by the Institute of Archaeology of Tatarstan Academy of Sciences (through Leonid Vyazov). James S. Williamson (Memorial University of Newfoundland, Canada) is kindly acknowledged for the English language editing of the manuscript. The authors are grateful for the constructive comments of two anonymous reviewers.

**Conflicts of Interest:** The authors declare no conflict of interest. The funders had no role in the design of the study; in the collection, analyses, or interpretation of data; in the writing of the manuscript, or in the decision to publish the results.

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
