# Peer review of "Shoreline Dynamics and Evaluation of Cultural Heritage Sites on the Shores of Large Reservoirs: Kuibyshev Reservoir, Russian Federation"

_water, doi:10.3390/w11030591_

Round 1

Reviewer 1 Report

The overall concept and rationale of the manuscript is fine. This is an interesting and pertinent problem, and as the authors correctly point out, an under-researched one. The scale of the problem in their study area is also impressive. I had never realised the sheer size of the reservoirs along the Volga and their environmental impact. However, there are some issues which prevent the paper from publication in its current form. Extensive revision is required to bring it to (in my opinion) international journal publication standard:

The main general issues are:

1. The rationale is there (i.e. generally under-researched, in need of local mitigation) but could be made stronger - for example, the local need is presented too far into the paper (pg. 5 actually). The motivation for the specific actions in the paper also needs to be explained better. For instance, will the reservoir rise or fall in future in response to climate change or future management practice and is this why you have chosen to test vulnerability by oscillating water levels? Why have you identified Beganchik as a case study? Is the erosion here representative of the wider situation or is it an extreme outlier? Or is it simply because it is the only remaining Pal/Meso site?

2. The methods section is not clear, and consequently it is not clear how the results are obtained and what their significance is and whether they are valid. For instance, I can see the following problems:

·         Overly generic statements: for instance, which remote sensing methods were used?

·         Lack of information: Which source was used to identify the reservoir outline in Figs 4-5?

·         Lack of information: How was the water level oscillation done (i.e. was a digital elevation model used)?

·         Uncritical spatial analysis: consideration needs to be given to how and when the archaeological database was compiled, as this affects how representative it is as a record.

·         Lack of information: How were shoreline retreat rate values obtained? Tools like DSAS or single representative measurements?

·         Unclear measurements: what do the specific land and volume indicators mean?

I recognize that there are language issues, but there simply is not enough information to allow the reader to understand what was done. To remedy this I suggest a clear division between the methods used for the reservoir-scale work and the Beganchik case study and more detail presented for each. Also, I suggest the authors look at some of the recent literature on coastal archaeological vulnerability and shoreline dynamics. This will a) show some of the techniques used and b) how the methods are written up. E.g. the following 3 recent papers might be helpful:

·         Andreou, G.M., Opitz, R., Manning, S.W., Fisher, K.D., Sewell, D.A., Georgiou, A. & Urban, T.. 2017. Integrated methods for understanding and monitoring the loss of coastal archaeological sites: the case of Tochni-Lakkia, Cyprus. Journal of Archaeological Science Reports 12: 197–208

·         O'Rourke, M.J.E. 2017. Archaeological site vulnerability modelling: the influence of high impact storm events on models of shoreline erosion in the western Canadian Arctic. Open Archaeology 3: 1–17. https://doi.org/10.1515/opar-2017-0001

·         Westley K (2018) Refining Broad-Scale Vulnerability Assessment of Coastal Archaeological Resources, Lough Foyle, Northern Ireland, The Journal of Island and Coastal Archaeology, DOI: 10.1080/15564894.2018.1435592

3. The English language of the text needs revision. I suggest the manuscript is passed by a native English speaker. This would greatly improve its readability and hence impact. I have indicated examples below where language needs to be changed. Note that this is not a full list of errors, but rather a representative sample of the most common errors.

4. Minor restructuring would help. Move the study area before the archaeological background, separate results and discussion. Clearly separate in the methods, results and discussion the reservoir-scale work and the Beganchik case study.

5. I question whether this journal and special issue are suitable for this manuscript. My feeling is that the shoreline dynamics work, while adequate for archaeology, is not sufficiently advanced for a volume on coastal processes. I think the article, if revised, would have more impact in an archaeology/heritage management journal. I leave this to the editors to decide if they feel if it fits in their issue.

Overall, I do think the manuscript is sufficiently interesting to warrant publication, but needs to be revised for the sake of clarity and scientific validity, and perhaps, aimed at a different journal.

Specific problems are:

Line 1-2: Title. English is not technically correct, specifically: “big reservoirs coastal areas”. The technical definition of a coast is the boundary where land meets the sea/ocean. Therefore this does not apply to reservoirs. A more appropriate term is ‘shore’. Thus this could be changed to “…from the shores of large reservoirs”

Lines 16, 27: ‘fast dynamic’ is incorrect English. ‘Highly dynamic’ would be more suitable

Line 27: ‘only left’ is incorrect English. ‘Last remaining’ would be more suitable

Line 32: ‘totalizing’ = incorrect English. ‘totalling’ would be more suitable

Line 57: Incorrect English, change to ‘The Volga River’

Line 61-62: Confusing sentence. Perhaps missing words? E.g. “due to the specific landscape in which they were built or constructed”

Line 67-68: sentence on climate change sits oddly. It does not link well to prior text. Be clear – what climate changes are likely to be detrimental in your study area? How might they affect the reservoir and this cultural heritage?

Line 69: ‘punctual’ = incorrect English (it means ‘on time’). Unclear from context what the correct term should be.

Line 71-71: Unclear what the link between sentences is. It goes from mention of past studies to integration of different methods. Need to be clearer if A) you are now referring to work in this study, and B) what these methods are.

Line 73: typo: scope NOT scopes

Line 73-77: this describes the work done in the paper, but needs a better link to the rationale (e.g. Line 66) which is that erosion risks along large inland water bodies are rarely considered. A little restructuring of these paragraphs would probably do it. Also – you need to be clear why you look at the Beganchik site. Is it to provide a localized case study to demonstrate both method and the problem? If so, please say so clearly.

Line 85: unnecessary text: delete “a number of”

Line 87: ‘in present’ = incorrect English. ‘at present’ is correct

Line 90-91: Unclear: are the PPCHP and Russia Cultural Heritage 2 the same projects?

Lines 90-100: direct relevance to this work is unclear, unless your study is part of it? If so, please state. If not, consider removing, it feels like filler.

Lines 101-107: Chronology. Please be consistent, you provide date ranges for some periods. Why not all? This is particularly the case for the Upper Pal, since it is such a long interval. E.g. does occupation go back to 40,000 BP or only to the final Upper Pal?

Lines 104-105: Incorrect English; suggested change: “The only remaining Palaeolithic site which has not yet been destroyed by the reservoir is Beganchik”

Line 109: Typo: the Volga-Bolgar

Line 124: unnecessary text: delete “a number of”

Line 142: Suggest restructuring so that the study area section comes before the archaeological background. It is easier for the unfamiliar reader to first discover where you are working and the wider context of reservoir construction before going into detail on the archaeology.

Line 145: punctuation: no need for () around region names.

Lines 148-151: English is poor, hence message is unclear

Line 152: incorrect English; use “area of interest”

Line 158-160: I think you need a bit more explanation of the terraces, since you mention them several times later in the text. It is not helped by poor English. E.g. suggested change – “Quaternary sediments make up the Volga-Kama terraces which formed in response to ….”

Lines 162-164: see previous comment re. use of term ‘coast’. Suggested alternatives: shore, shoreline, bank

Lines 163-165: Statements like this re. the need for this work need to come in the introduction to the paper, as they would greatly strengthen the motivation for the research.

Line 172: Which modern remote sensing methods do you use? You cite Bucci et al. 2018 as a supporting reference but there is no evidence that you employ the same underwater remote sensing techniques as they do. I suggest you specifically mention which techniques you use and find more appropriate supporting references.

Line 183: Point density tool: what parameter did you use – e.g. search radius?

Line 183-85: Be clear: How did you adjust level the water level? Did you raise/lower water levels over a digital elevation model? If so, which one (you do not mention one in the previous section)? Why the values of +/-0.5 to 1m chosen? Do they relate to future climate predictions or are they arbitrary?

Line 195: Results and discussion should be separate sections. Results should simply present the outcomes of the analysis (e.g. shoreline retreats at rates of xxxxx, xxxx sites are at risk); discussion should then discuss the implication and significance of the results.

Line 242: More information is needed on the archaeological database. How was it compiled – e.g. field survey, remote sensing, development-led? When was it compiled – e.g. pre- or post-reservoir? I ask these questions because, if you do spatial analysis, you need to be more critical of the patterns shown by the data. For example, you identify concentrations of sites, but don’t identify if these areas have been surveyed more systematically than areas with few sites. Also, Figure 3 suggests many of the sites are located on the edge of the reservoir. Is this because they were identified after reservoir construction? The implication would be that many undiscovered sites could be underwater inside the reservoir.

Figure 6: Legend missing

Line 310: What is your analysis based on? Points presumably. However, bear in mind that single points are poor representations of large sites – i.e. they cover a larger extent than indicated by a point, hence your estimate of vulnerability may be conservative.

Tables 4 and onwards: Please explain the specific land and volume loss indicators. I have not seen these before. You also need to define the vulnerability categories – e.g. what makes a particular rate extremely dangerous? Also – use . rather than , to indicate decimal places. How are you getting these erosion rates? Are they averages along the shore based on multiple transects generated by software like DSAS, or are you picking them from, single representative transects?

Line 367: typo: waves

Line 372: be specific – what are ecosystem-based mitigation measures?

Section 5.4: I have to confess to being a little confused re. Beganchik. It looks like an island in Fig 7, but a peninsula in Fig. 8 and on Google Earth? There is clear erosion on its northern and eastern sides, but has there also been accretion on its southwest side? If so, you need to mention this too – it highlights the complexity of shoreline dynamics in that some areas will prograde and others erode.

Author Response

Dear Reviewer,

Many thanks for your helpful comments and suggestions. Point-by-point responses are highlighted in the attached file in red colour.

Reviewer 2 Report

The manuscript, in general, has a good idea: to point out the necessity for improved management and protection of cultural heritage sites at coastal areas of big river’s reservoirs. That motivation and moral of the story told here are important and should be treated as such, specifically since this kind of research is not so presented now.

However, there are some serious issues with the structure and communication of the message mentioned above to the readers and they are listed below:

-    The title is a little confusing – the expression“…sites from big reservoirs coastal areas…” is wordy and it is not clear what is the “assessment” about? It may be flooding risk assessment, erosion risk assessment, etc.

-    Furthermore, Authors imply that due to the lack of studies dealing with coastal erosions in reservoirs the interest of the manuscript is to deliver such analysis by analysing the changes in Volga River and then different data representation of historic sites. This is then followed by a huge and detailed description of multiple heritage sites for 4 different sectors in the case study region. This part is dense to read, and a lot of paper is “spent” on it, while there are big gaps in technical descriptions of the used methodology, tools, criteria for analysis, statistical data.

-    In the Materials and Methods section, Authors list the used equipment, data and software. This part should be explained in much more detailed way, e.g. which data was used from which source (old maps, existing GIS systems, UAV), what information was used from drone flights – only photos I presume? If they did what anomaly detection algorithms have been used to apply these data in the model? What is the model here anyway?

-    What do these “working scenarios” mentioned at page 5, line 184 really take into an account?  They are mentioned as a step in methodology, but there is no indication of the setup in which they were used and the final conclusion regarding these so-called scenarios on page 1 again (the page numeration is messed up from 10th page onward), line 310-316, just before figure 7, the conclusion is given that there is no effect of water oscillations up to +-1m. Although there is a whole paragraph describing the Volga River dynamics, nowhere is mentioned the river flow and its variations during all these periods and the same thing goes for river velocities which are together an inevitable part of “river dynamics”.

-    There should be some clear differentiation between the first sectorization of higher order (figure 4) regarding the whole area and the second sectorization which is focused on Beganchik site only. It is not clear what do “working scenarios” have to do with this specific site if it is relatable anyways.

-    Finally, this last focus on the Beganchik case study is also very vaguely interpreted; how is the recent drone + ground-truthing survey data overlapped with historic maps to have comparison delivered by tables 4-6? When analysing the coastal erosion processes here, one should not omit the effects of the wind, waves and river velocities, and there are no relevant data (for waves it is understandable, but winds at least?). How are these effects taken into the account or if they were not, on which assumptions was that based? Finally, why is the legend for the level of danger apparently related only to the height of the coast? There are no clearly defined criteria for characterization of some site to be dangerous – that should be clarified in the manuscript.

-    The conclusions taken from the tables 4-5 are implying only the great effect of erosion in the first two periods with regards to the others, but that is obsolete since these two periods are simply longer causing the erosion to have more time reducing the coast.

-    Why should mitigation measures be focused on sector 3 of Beganchik site, when it has the least danger of them all, and what exact measures would those be? Also, why is there (sector 3, table 6) higher shoreline retreat in the period 2008-2014 than it was before in a longer period of 22yrs, Authors should discuss this deviation.

The literature background and previous studies are presented very thoroughly and with a great level of details.

Most of the major concerns are given above, while the few points are emphasized by highlights and comments in the manuscript itself. 

In conclusion, the way the paper was written implies that the research is more like a detailed review of the history in the Kuibyshev reservoir, but it is obvious that some technical methodology was implemented and used. Therefore, in my honest opinion, the manuscript should address these technical issues and have well defined the steps and algorithms used in the methodology to be published.

Author Response

(The authors gave the same response as above.)

Round 2

Reviewer 1 Report

Overall, I am largely satisfied with the changes made by the authors. The justification is better, the method is clearer and the new structure is easier to follow. I am therefore happy to recommend publication in Water, with the caveat of a few minor correction.

More specifically, these are:

1.       The English is generally readable, but there are some improvements could be made: For instance in the abstract use of the term ‘high dynamic’ (itself a replacement for the original ‘fast dynamic) is still incorrect. Suggested changes are: Line 15: new shorelines, which are highly dynamic regarding….Line 27: results show that the coastal area is highly dynamic….Please run thorough the rest of the paper and have a good check for similar errors.

2.       Line 67: Use of the term ‘destroyed’ in relation to archaeological sites and reservoir creation. This is because inundation does not necessarily always destroy sites. If it did, there would be no sub-discipline of submerged prehistory. It’s actually possible that you have many sites preserved underwater in the reservoir (which is really interesting but another story altogether...). Therefore, a suggested alternative and less judgemental term is ‘impacted’. Please take a look through the rest of the manuscript and see where else this might have to be edited.

3.       Justification is definitely better – but please make clear why Beganchik is used closer to the start. E.g. integrate Lines 199-200 with the final paragraph of the Introduction. Also, explicit state before line Line 69 that you include a detailed case study to demonstrate the destructive potential of wave erosion and show how monitoring can be accomplished. This will nicely set up your use of the Beganchik case study.

4.       Lines 168 to 174. Great that the DEM is mentioned, but you still need to indicate that your choice of +/-0.5 to 1m water level is arbitrary. You can justify this by including the info you provided in your response re. the controls on water level– i.e. briefly highlight the controls on water level and then state that your large-scale study will showcase the impact that minor oscillations in water level can have by virtue of arbitrarily increasing/decreasing it by +/- 0.5 to 1m

5.       I accept that there are deficiencies with the archaeological database – this is not uncommon. But I think you could include a line somewhere between Lines 156-164 which acknowledges this. E.g. survey has been unsystematic across the study area, sites are located to varying degrees of accuracy and full extent of individual sites is not necessarily know. However, it is the most comprehensive archaeological dataset which currently exists for this area, hence will be used for this study.

6.       Happy to see more discussion of your results – but please signpost things better for your reader. E.g. Line 398: state table numbers, rather than just saying ‘in the tables’. Also be clear which Sectors you are talking about since you have Sectors 1,2,3 for both Beganchik and the large-scale study.

Author Response

Dear reviewer,

We really appreciate your comments and suggestions. In the attached file you will find the suggested comments and our answers.

Reviewer 2 Report

The manuscript has been significantly improved, especially with regards to the description of the used methodology and tools. The structure is also improved, along with some data representation.

I am a little bit concerned for the lack of meteorological and hydrological data, but considering the restricted data availability that Authors have encountered, they have done remarkable work and should be published in Water.

Few additional notes that would be preferable to address prior to the publication:

- The new title is better and more evident. However, only the preposition "from the shores" should be "on or at the shores", otherwise "from the shore" implies some distance from the shore.

- All in all English is not bad, and the writing style is clear and concise -however, moderate changes (whether to check it with English native speaker of official editing service) should be done; e.g. preposition in title, or was/were for plural on page 5 line 177 (should be "were used")  

- One of my previous comment was regarding the dangerous criteria in the last part of the result section, and it was not addressed properly. It is not clear from the manuscript what defines the danger? Is it the relation between the height of the coast and the level of corresponding erosion, if so, it should be clarified in the manuscript? Also, if that is the case – and the height of coast is an indicator for erosion danger, why is the sector 2, which is said to be moderately dangerous (pg 11, line 366), also said to be the most exposed and threatened by erosion (pg 11, line 359)?

Author Response

Dear reviewer, 

We appreciate your suggestions and comments for the early version of the manuscript. You also have attached the answers for the minor revision.

Kind regards.
